# The Virucidal Effect of the Chlorination of Water at the Initial Phase of Disinfection May Be Underestimated If Contact Time Calculations Are Used

**DOI:** 10.3390/pathogens12101216

**Published:** 2023-10-03

**Authors:** Fredy Saguti, Inger Kjellberg, Marianela Patzi Churqui, Hao Wang, Timur Tunovic, Jakob Ottoson, Olof Bergstedt, Helene Norder, Kristina Nyström

**Affiliations:** 1Institute of Biomedicine, Department of Infectious Diseases, University of Gothenburg, 413 46 Gothenburg, Sweden; 2Göteborgs Stad Kretslopp och Vatten, 424 23 Gothenburg, Sweden; 3Department of Risk and Benefit Assessment, Swedish Food Agency, 75126 Uppsala, Sweden

**Keywords:** chlorine, chlorine dioxide, virus viability, disinfection

## Abstract

For the microbiological safety of drinking water, disinfection methods are used to remove or inactivate microorganisms. Chlorine and chlorine dioxide are often used as disinfectants in drinking water treatment plants (DWTPs). We investigated the effectiveness of these chemicals in inactivate echovirus 30 (E30), simian 11 rotavirus (RV SA11), and human adenovirus type 2 (HAdV2) in purified water from a DWTP. Within two minutes of contact, chlorine dioxide inactivated E30 by 4-log_10_, RV SA11 by 3-log_10_, and HAdV2 could not be detected, while chlorine reduced E30 by 3-log_10_, RV SA11 by 2–3log_10_, and HAdV2 by 3–4log_10_. However, viral genomes could be detected for up to 2 h using qPCR. The CT method, based on a combination of disinfectant concentration and contact time, during such a short initial phase, is problematic. The high concentrations of disinfectant needed to neutralize organic matter may have a strong immediate effect on virus viability. This may lead to the underestimation of disinfection and overdosing of disinfectants in water with organic contamination. These results are useful for the selection of disinfection systems for reuse of treated wastewater and in the risk assessment of water treatment processes using chlorine and chlorine dioxide.

## 1. Introduction

In the 19th century, the effect of disinfectants, such as chlorine, was discovered. It was first used in medical wards, but later also for water disinfection [1]. The chlorine used for the disinfection of drinking water was liquid chlorine stored under pressure in cylinders. At present, sodium hypochlorite (NaOCl) is often used since it is easier to handle compared to liquid chlorine.

Chlorine has a virucidal effect, the mechanism of which is not fully understood and is debated. The high chemical reactivity of chlorine may inactivate viruses by replacing hydrogen atoms with chlorine atoms, thereby degrading the viral nucleic acid, while often leaving the virion structure largely unchanged [2]. Chlorine also reacts with organic matter in the water, which results in disinfectant by-products (DBPs), such as trihalomethanes (THM) and halogenated acetic acids (HAA). The disinfection efficacy of chlorine is highly pH and temperature-dependent. Its disinfection efficacy decreases with increasing pH and decreasing temperature [3]. 

Another chlorine compound used for disinfection is chlorine dioxide. It is about 10 times more soluble than chlorine in cold water. It acts as a free radical by reacting strongly with reducing agents and attacking electron-rich organic molecules with a low production of THMs [4]. Chlorine dioxide is effective at a broad pH range, between pH 5 and 10, however, it is unstable [4] and is typically produced on site. Even at low concentrations in water, it has potent antimicrobial activity against many microbes, including viruses [5,6,7]. Like free chlorine, chlorine dioxide reacts with the genomes of viruses, and can also cause structural changes to the surface of enveloped viruses [8,9].

The effectiveness of inactivation of pathogens by disinfectants is measured by a combination of chlorine concentration and contact time (CT). “CT” refers to the product of chlorine concentration (C) and contact time (T); these metrics are used to determine whether the treatment has been effective in reducing the concentration of harmful microorganisms in the water [10]. The CT method is often used in municipal water treatment systems to ensure that the water is safe to drink. 

Several studies have compared the inactivation efficacies of both chlorine and chlorine dioxide against enteric viruses with inconsistent results [6,11,12,13,14]. The divergent results of the various studies were associated with the pH, temperature, and ionic strength of the investigated water as well as the amount of added disinfectant [15]. In addition to those factors, preparation of test microorganisms can significantly affect the inactivation efficiency of the disinfectant tested in bench scale experiments. Studies have shown that culture medium or aggregated viruses in the analysed water can both affect virus resistance to free chlorine by absorbing free chlorine [16,17]. 

Enteric viruses are important waterborne pathogens reported to cause outbreaks worldwide with severe illness and death, especially to elderly people and children under 5 years of age [18,19]. Normally, these viruses are shed at high numbers in faeces and are reasonably stable in the environment [20,21,22]. They can contaminate source water and drinking water and can ultimately be consumed by the public [20,21,22,23]. Drinking water treatment plants (DWTPs) need to adopt stringent disinfection regulations for the inactivation of enteric viruses during the purification steps. Several microbial barriers are used for removal or inactivation of microorganisms including membrane filtration and UV-radiation. However, relatively high doses of UV are required for the inactivation of viruses [24]. Ultrafiltration (UF) membranes with a smaller pore size than the target virus can be used to achieve a high virus reduction, but even UF does not completely eliminate viruses from the water [25]. 

The U.S. Environmental Protection Agency (EPA) has published the drinking water Contaminant Candidate List 4 (CCL4), which includes enteric viruses such as adenovirus and echovirus [26]. According to the US National Primary Drinking Water Standards, enteric viruses must be inactivated by 4-log_10_ [27]. The required inactivation in log reduction in a microorganism is often estimated by determining the CT value as a proxy for viral inactivation, though CT does not measure viral inactivation. The U.S. EPA Guidance Manual suggests that a CT value range of 25.1–33.4 mg·min/L for chlorine dioxide and 6–8 mg·min/L for free chlorine is required at temperatures of 5–10 °C and pH values of 6–9 to achieve a 4-log_10_ viral reduction [28]. The European Commission’s drinking water directive requires that drinking water be free of substances or microorganisms that can affect the health of consumers [29]. Swedish drinking water treatment plants follow this directive and use the Scandinavian model for microbial barrier analysis guidance, which demands the use of good raw surface water quality and a virus reduction of at least 4-log_10_ and up to 6-log_10_, depending on faecal contamination of the raw water, to be achieved by DWTPs [30].

In this study, the chlorine and chlorine dioxide inactivation of three pathogenic viruses was determined in a simulation of the conditions at a DWTP during episodes with high organic content, which demands high initial doses of the disinfectants to achieve their measurable concentrations after two hours of contact time. 

## 2. Materials and Methods

### 2.1. The Drinking Water Treatment Plant

Lackarebäck DWTP is one of the two plants that supply drinking water in Gothenburg, Sweden. The plant uses both conventional treatment steps, including coagulation, sedimentation, rapid filtration, and chemical disinfection, as well as ultrafiltration of the water with filters of nominal pore size of 20 nm. The chemical disinfection process involves a combination of chlorine and chlorine dioxide in the drinking water reservoir as single-contact tanks. Chlorine and chlorine dioxide concentrations after the mixing chamber and reservoirs are used for calculating concentration. The effective contact time is calculated by adjusting the total time with a factor of three, due to the hydraulics, e.g., it is the effective contact time for the first 33% of the water passing the contact tanks. The resulting CT values, pH, and temperature are then used to determine the log reduction achieved.

### 2.2. Chlorine and Chlorine Dioxide Stock Solution Preparation

The chlorine stock solution used was 12% sodium hypochlorite (NaOCl) in aqueous solution, equivalent to 150 g/L Cl_2_ (Univar Solution Inc., Downers Grove, IL, USA). This solution was added to 4 L of the ultrafiltrated drinking water to be analysed. The concentration of chlorine was measured according to the SS-EN ISO 7393-2 method. Chlorine dioxide was produced in a reactor at the plant on the day of the experiment by mixing hydrochloric acid (HCl) and sodium chlorite (NaClO_2_).

The chlorine dioxide concentration was determined according to the SS-EN 1031-4 method using Ion chromatograph 930 Compact IC Flex (Metrohm AG, Herisau, Switzerland), following the EPA method 300.1 and OSHA ID-202 protocol. Total chlorine and free chlorine concentrations in sampling water were measured at each sampling time point, 0 min, 2 min, 10 min, 30 min, 60 min, and 120 min using the SL1000 Portable Parallel Analyser with free chlorine and total chlorine Chemkeys (HACH, Loveland, CO, USA).

### 2.3. Assessing the Effect of Cell Culture Media on the Chlorine and Chlorine Dioxide Concentration in Ultrafiltrated Water

The required inactivation in log reduction in a microorganism is often estimated by determining the CT value. The disinfectants residual concentration at each of the five selected sampling points were used for calculating the CT values. To reach the required CT value and measurable disinfectant concentrations over 2 h, the amount of chlorine and/or chlorine dioxide had to be adjusted to the content of the solution to be disinfected.

Cell media used for virus cultivation were tested for chlorine adsorption at dilutions of 1/100 and 1/1000 in the UF water from the Lackarebäck DWTP. Larger particles in the cell culture media were removed using 0.45 um filters. The cell culture media dilutions were treated with different concentrations of chlorine and chlorine dioxide after filtration. The residual oxidant levels of free chlorine and chlorine dioxide were monitored for up to 120 min of contact time to obtain a final residual concentration of at least 0.03 mg/L (Figure 1; Appendix A).

### 2.4. Host Cells and Virus Stocks

Stocks of echovirus 30 (E30) and simian rotavirus (RV SA11) were obtained from the American Type Culture Collection (ATCC, Manassas, VI, USA), and human adenovirus 2 (HAdV2) was obtained from Dr Wang [31]. E30, RV SA11, and HAdV2 were propagated in African green monkey kidney cells (Vero CCL-81), African green monkey foetal kidney cells (MA-104), and human lung carcinoma cells (A549 cells) (ATCC, Manassas, VI, USA), respectively. Vero and A549 cells were grown in Minimum Essential Media (MEM; Gibco, Bleiswijk, The Netherlands) supplemented with 5% foetal calf serum (FCS), 1% L-glutamine, 100 units/mL of penicillin, and 100 μg/mL of streptomycin (Pen-Strep; Gibco, Life Technologies Corporation, Grand Island, NY, USA). MA-104 was grown similarly but in medium 199 (M2154; Sigma Aldrich, St. Louis, MO, USA) with 2.5% FCS, as previously described [24]. 

### 2.5. Viral Infectivity Titres

Virus titres were determined by median tissue culture infectious dose (TCID_50_). The 10-fold serial dilutions (10^−1^ to 10^−7^) of the viruses were prepared and inoculated in 10 replicates per dilution on ~80% confluent monolayer of the same cell lines used for propagation in 96-well plates (Nucleon Delta Surface, Thermo Scientific, Roskilde, Denmark). The plates were incubated at 37 °C in a humidified 5% CO_2_ atmosphere and observed daily for 4–9 days until cytopathogenic effects were observed. The TCID_50_ titres of virus were determined when 50% of the cell cultures in wells showed full CPE [32]. The number of viral particles per litre could be estimated by assuming that one millilitre TCID_50_ will produce 0.69 plaque forming units (PFUs)/mL [33]. The initial titres, TCID_50_/mL, of E30 was 1 × 10^7^, the RV SA11 titre was 5.62 × 10^7^, and HAdV2 was 1 × 10^8^.

### 2.6. Chlorine and Chlorine Dioxide Treatment of Water Samples Containing Viruses

The ultrafiltrated water samples used were collected on the day of the experiment and all relevant parameters such as temperature, pH, and water quality (by measuring of transmission at 254 nm) were determined prior to each experiment (Appendix A). 

The effect of chlorine and chlorine dioxide on the viability of E30, RV SA11, and HAdV2 was investigated by adding virus solutions to water treated with chlorine or chlorine dioxide. Briefly, 5 mL of virus with a TCID_50_ between 10^7^ and 10^8^ in cell medium was added to 5 L of ultrafiltrated water and mixed well to obtain an initial virus concentration of 10^4^ to 10^5^ TCID_50_/mL, which can detect a 4-log_10_ viral inactivation. Before adding the disinfectant solution into the flask, 1 L of the solution was withdrawn as untreated virus control. Disinfectant stock solution was then added to the 4 L virus solution to generate an initial disinfectant dose of 0.5 mg/L ClO_2_ or 1.3 mg/L Cl_2_, which was stirred throughout the process. At each selected time point (2 min, 10 min, 30 min, 60 min, and 120 min), 10 mL of treated and untreated virus solution samples were collected. The treated virus samples were immediately neutralized by adding 0.01 mL of sodium thiosulfate to obtain a final concentration of 0.1 M as to stop the disinfection and preclude the effects of residual oxidants.

To investigate the viability of the viruses after disinfection at different time intervals, treated and untreated samples were serially diluted (10^−1^ to 10^−7^) in cell culture medium and inoculated in 10 replicates per dilution on ~80% confluent monolayer of the same cell lines used for propagation of the viruses in 96-well plates. The plates were incubated at 37 °C and monitored daily to detect any CPE. The viral titres were determined as TCID_50_/mL.

### 2.7. qPCR Determination of the Virus in the Cell Culture

To confirm the presence of the virus in both control and treated samples, real-time qPCR analysis was performed on the first dilution after cell culture of all treated and untreated water samples. Briefly, total nucleic acids were extracted from 300 µL of mixed cell cultured solution of each three viruses from both control and treated samples with disinfectants in Maxwell RSC instrument AS4500 (Promega, Madison, WI, USA) using Maxwell RSC Viral Total Nucleic Acid Purification kit (Promega, WI, USA) according to the manufacturer’s instructions. Viral nucleic acids were eluted with 50 µL elution buffer. 

All samples were tested in triplicate. The reaction for RNA viruses (E30 and RV SA11) was performed in a 25 µL reaction mixture containing 5 µL of the extracted nucleic acids, 1 × Reaction Mix (Invitrogen, Carlsbad, CA, USA), 20 U RNaseOUT TM (Invitrogen, Carlsbad, CA, USA), 0.5 µL SuperScript^TM^ III/platinum^TM^ Taq Mix (Invitrogen, Carlsbad, CA, USA), 0.4 µM of each primer, 0.2 µM of probe, and 4 µL of water. The qPCR was initiated with reverse transcription at 50 °C for 30 min followed by one cycle of 95 °C for 10 min and 45 cycles of 95 °C for 15 sec and 55 °C 1 min. The reaction mixture for HAdV2 was performed in a 20 µL reaction containing 2 µL of DNA, 1 × TaqMan Universal PCR Master Mix (ThermoFisher, Waltham, MA, USA), 0.5 µM of each primer, and 0.4 µM of probe and 5.2 µL of water. The qPCR was initiated with a cycle of 50 °C for 2 min and 95 °C for 10 min, followed by 45 cycles of 95 °C for 15 sec and 55 °C for 1 min, followed by the extension cycle of 60 °C for 1 min. The sequences of primers and probes used are listed in Appendix A. Four tenfold serial dilutions (1/10^5^–1/10^8^) of a 2 µg plasmid containing all targeted regions of virus genomes inserted into EcoRV site of pUC157 plasmid (pUC57cl; GenScript HK, Ltd., Hong Kong, China) were used as a positive control in all qPCR. Sterile water (Sigma Life Science, Darmstadt, Germany) was used as the negative control. The detection of each virus by qPCR was performed using either the 7300 Fast Real-Time PCR or the QuantStudio™ 5 (QS5) Real-Time PCR instruments from (Applied Biosystems, Foster City, CA, USA).

### 2.8. Calculation of CT Values

Exposure to residual disinfectant concentrations were quantified in terms of CT values calculated in Equation (1), using the solver function in Excel 2016 (Microsoft Corp., Redmond, WA, USA). Free chlorine and chlorine dioxide concentrations were measured at each sampling time point, 0 min, 2 min, 10 min, 30 min, 60 min, and 120 min.
(1)$$\mathrm{CT}=\sum_{i}^{n}[((c_{2}-c_{1})/2)*\left(t_{2}-t_{1}\right)]$$
where $c_{1}$ is the initial concentration at $t_{1}$ and $c_{2}$ is the residual concentration at time $t_{2}$. $\sum_{i}^{n}$ is summation of calculated CT of sampling points, *i* is initial sampling point, and *n* is a designated end of sampling point. The time variable t is the time at the selected sampling time points.

## 3. Results

### 3.1. Testing of Cell Culture Media for Chlorine Consumption

Initial consumption of chlorine by the organic compounds in the two cell culture media was investigated prior to analysing the inactivation efficiency of viruses. The media were tested in dilutions of 1/100 and 1/1000. There was a rapid decrease in free chlorine residuals within the first 2 min in both cell culture media. Due to the rapid consumption, the media were in addition filtrated through 0.45 um filters to remove larger debris. The chlorine adsorption declined but was still high at a dilution of 1/1000 and 0.45 um filtration (Figure 1A–C). Due to this rapid consumption, the initial dose of each disinfectant had to be determined to achieve measurable residual concentration after 120 min in the water, as is the routine at the DWTP. An initial chlorine concentration of 1.30 mg/L was required in order to obtain the desired final concentration of residual chlorine at 0.03 mg/L. For chlorine dioxide, an initial concentration of 0.45 mg/L was needed to obtain the required concentration of 0.03 mg/L, as shown in Figure 1D.

### 3.2. Inactivation of Viruses by Chlorine and Chlorine Dioxide

To achieve viruses with sufficiently high titres to obtain a log_10_ reduction of 4 or more in 5 L of water, each virus was passaged several times in cell cultures to reach as high a titre as possible. The titre of the diluted viruses in treated and non-treated water samples are given in Table 1.

The concentrations of chlorine dioxide and free chlorine at the experiments with chlorine were measured at the same time points during virus inactivation. The first possible time point to measure chlorine dioxide and free chlorine was after two minutes when a high consumption had already occurred. Thereafter measurements were performed at an additional four time points from the start of each experiment (Table 1). The average CT values in this study for chlorine were 1.5–1.8 mg·min/L and for chlorine dioxide were 0.80–0.85 mg·min/L (Table 2, Table 3 and Table 4). These CT values were used according to the Scandinavian method [30] to calculate the expected reduction resulting in expected virus log_10_ reductions of 1.2 for chlorine and 0.25 for chlorine dioxide. The achieved virus reduction during the first two minutes was thus approximately three times higher than expected with chlorine and more than ten times higher than the expected with chlorine dioxide.

The detection of E30, RV SA11, and HAdV2 were performed using RT-qPCR and qPCR assays on both untreated and treated samples with Cl_2_ and ClO_2_. The viral genomes of all three viruses were able to be amplified even when viability of virus in cell culture was negative (Table 1).

### 3.3. Echovirus 30 (E30)

In the E30 suspensions with minimum essential medium (MEM), an initial dose of 1.31 mg/L chlorine and 0.53 mg/L chlorine dioxide in 4 L water was required to obtain a chlorine residual of 0.11–0.12 mg/L free chlorine and 0.07–0.09 mg/L chlorine dioxide after 120 min (Appendix A). The virus inactivation occurred very rapidly by the disinfectant dosages. Already after two minutes of contact time, the viral reduction was >3.0 log_10_ after chlorine treatment and between 3.25 log_10_ and 3.75 log_10_ after treatment with chlorine dioxide (Table 1). The concentration of free chlorine residuals was 0.37–0.50 mg/L at 2 min, with a CT value of 0.79–1.81 mg·min/L. The chlorine dioxide residual of 0.20–0.26 mg/L at 2 min achieved a CT value of 0.73–0.79 mg·min/L (Table 2). Viral RNA and viable virus could be detected after two minutes of treatment. However, viral RNA was also repeatedly detected in one sample each after each treatment with chlorine dioxide (after 60 min) and chlorine (after 30 min) without detection of viable viruses in the cell culture (Table 1).

### 3.4. Rotavirus SA11 (RV SA11)

For RV SA11 in M199 medium dispersed in water samples, an initial dose of 1.13–1.20 mg/L chlorine and 0.49–0.52 mg/L chlorine dioxide in 4 L water resulted in a chlorine residual of 0.10–0.11 mg/L free chlorine and 0.03–0.06 mg/L chlorine dioxide after 120 min of contact time (Table 3). The virus was detected after two minutes contact time with both chlorine and chlorine dioxide in titres between 1.25 and 2.25 log_10_ TCID_50_/mL. The reduction after two minutes was between 2.75 log_10_ and 3.75 log_10_ (Table 1). Chlorine residual of 0.31–0.41 mg/L at 2 min gave a CT value of 1.44–1.61 mg·min/L. For chlorine dioxide, the viral reduction was between 3.50 and 3.75 log_10_. The chlorine dioxide residual of 0.16–0.25 mg/L at 2 min achieved a CT value of 0.8 mg·min/L (Table 3). The RT-qPCR detected RV SA11 genomes even after 120 min of contact time with either of disinfectants.

### 3.5. Human Adenovirus 2 (HAdV2)

HAdV2 grown on A549 cells in MEM medium required an initial dose of 1.31 mg/L chlorine and 0.47–0.56 mg/L chlorine dioxide in 4 L water in order to obtain a chlorine residual of 0.08–0.13 mg/L free chlorine and 0.03–0.08 mg/L chlorine dioxide after 120 min (Appendix A). HAdV2 was rapidly inactivated and achieved 3–4 log_10_ reduction after a contact time of 2 min (Table 1). Viable HAdV2 viruses were detected by cell culture assay, in one out of two samples, after being treated with chlorine for 2 min. Free chlorine of 0.35–0.55 mg/L at 2 min achieved a CT value of 1.66–1.86 mg·min/L. The chlorine dioxide residual of 0.25–0.35 mg/L at 2 min achieved a CT value of 0.85 mg·min/L (Table 4). Adenovirus DNA could be detected by qPCR after 120 min contact time with chlorine or chlorine dioxide, without detecting viable viruses.

## 4. Discussion

A substantial reduction in viability of 3-log or more was achieved for E30, RV SA11, and HAdV2 within the first two minutes of contact time with either chlorine or chlorine dioxide. No viable virus was observed beyond this time point. Due to the organic matter in the cell medium used for cultivation of the viruses analysed, we needed to use a relatively high initial concentration of chlorine and chlorine dioxide to obtain a measurable final concentration of residual chlorine after 120 min of effective contact time. Although this high dosage was required to quench the organic matter in the analysed water, these chlorine doses were in the WHO-recommended range of 0.2–2.0 mg/L for water treatment [34,35]. The high difference between the observed viral reduction in this study and by the CT method expected reduction in the viruses [30] indicates that the CT method can cause major overdosing of chlorine and even more for chlorine dioxide. Overdosing results in unnecessary environmental impact from CO_2_ emissions and energy use when the chemicals are produced. Overdosing can also affect consumer acceptability because of odour. Chlorine is also known to produce disinfection products which are harmful to human health. The finding in this study with the need of initially high doses of disinfectants in waters with slight contamination of organic matter may also have implications on the new need to reuse greywater as well as wastewater [36,37]. 

A previous study [38] performed with water from the same DWTP using three different phages showed a similar actual high initial log_10_ reduction compared to expected for CT calculations with chlorine dioxide. The difference between chlorine dioxide and chlorine is consistent with our study. For chlorine, lower doses than in this study were used and the phages which were selected for their chlorine resistance, explaining the difference. It is worth noting that the amount of free chlorine and chlorine dioxide residuals measured after 2 min of contact time in this study were relatively similar to residual concentrations measured in other studies, irrespective of the amount of disinfectant dose added at the initial phase [12,16,39,40]. 

In this study, the efficiency of chlorine dioxide in inactivating viruses was similar to that obtained by chlorine. A previous study observed no difference between the disinfection efficiency of chlorine and chlorine dioxide used against murine norovirus [11], which is consistent with our results. This is in contrast to other studies showing a higher efficiency by chlorine dioxide after one minute of contact time [12,41]. In this study, relatively lower amounts of chlorine dioxide compared to free chlorine were consumed by both ultrafiltered water and water with organic matter from diluted cell medium. Consistent with that, chlorine dioxide has been shown to be less sensitive to organic contaminants in water, and also more biocidal over a wide temperature and pH range compared to chlorine [39,40,42]. Despite all the advantages of chlorine dioxide over chlorine, it is unstable, which prevents its storage. Therefore, it must be manufactured on site before being added to the water, making it more difficult to use and handle than chlorine.

Apart from the high initial chlorine concentration, temperature and pH also have an effect on the inactivation of viruses, with increasing inactivation rate at higher temperatures [16,40]. Previously, it was shown that chlorine and chlorine dioxide inactivation efficiency are pH-dependent, especially when lower doses are used for viral inactivation [3,14]. With the high chlorine doses, RV SA11 inactivation was shown to be independent of pH level [14].

Despite the fact that viable viruses could not be detected after two minutes of treatment by cell culture assay, our results showed that qPCR and RT-qPCR assays could detect the viruses for up to two hours. However, the qPCR does not distinguish between viable and inactivated viruses, since it only detects the viral genomes. Similar results have been reported when Cl_2_- or ClO_2_- inactivated murine norovirus could be detected by RT-PCR [11]. The mechanism of inactivation by free chlorine of adenovirus was shown to occur between the virus attachment to the host cell and prior to viral protein synthesis [43]. This suggests that qPCR and RT-qPCR techniques are highly sensitive for detecting viral genomes that have even undergone up to 4-log of inactivation. However, the qPCR and RT-qPCR techniques do not provide information on virus viability after exposure to chlorine and chlorine dioxide. This can only be determined by virus cultivation in cell culture or infection of animals.

Although we designed a rather large bench-scale study, adjusting chlorine adsorption by organic matter in cell media, this study has some limitations. In a full-scale at DWTP, chlorination is a continuous process. Due to the large study design, we could not analyse viral inactivation before two minutes of treatment, since the chlorine and chlorine dioxide needed time to properly mix before sampling, just as in full-scale studies, and their concentrations could only be determined after two minutes of mixing. Early sampling points shorter than two minutes after addition of the disinfectants could have provided additional data on the viral inactivation, especially for the viruses showing complete early inactivation by either disinfectant.

## 5. Conclusions

In conclusion, benchmark experiments should be carried out to simulate actual environmental conditions at DWTPs for virus inactivation. We observed that HAdV2, RV SA11, and E30 were readily inactivated in ultrafiltrated water from Lackarebäck DWTP with a high initial concentration of either disinfectant used. The water contained a cell culture medium, and our results indicate that multiple sets of modernized water barriers should be used in removing organic matter impurities prior to the chlorine disinfection step. More studies should be conducted on partially treated waters to determine the inactivation efficiency of different viruses using chlorine and chlorine dioxide concentrations normally used at DWTPs to ensure the safety of drinking water without overdosing. There is also a need to investigate the inactivation efficiency of these disinfectants in water containing organic matter as greywater or wastewater to be reused to ensure their safety. 

## Figures and Tables

**Figure 1 pathogens-12-01216-f001:**
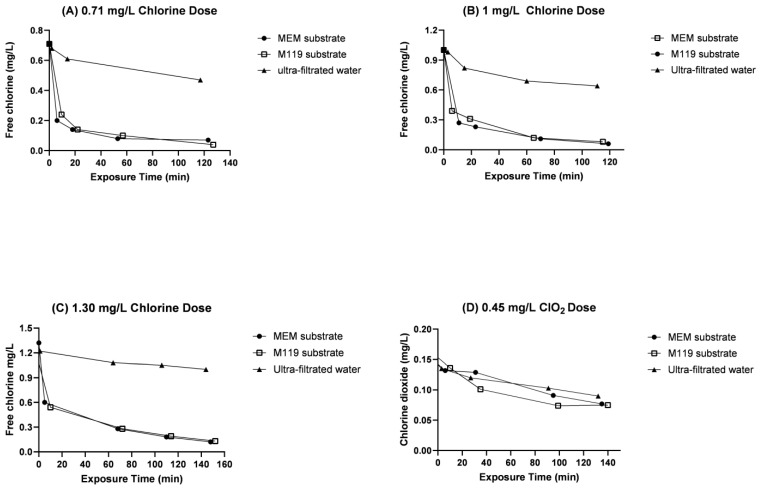
Test of internalization for the show free chlorine residuals after exposure to both MEM substrate and M199 substrate. The chlorine reactions were tested in three doses 0.7, 1.0, and 1.30 mg/L in (**A**–**C**). Chlorine dioxide dose tested was 0.45 mg/L with a contact time of 120 min (**D**).

**Table 1 pathogens-12-01216-t001:** Virus titres obtained in the treated and untreated water samples at each tested time point in relation to qPCR Ct values of the cell media after inoculation of the water samples.

Virus	Time (min)	Chlorine				Chlorine Dioxide			
		Not Treated Virus Log_10_ TCID_50_/mL (qPCR Ct Value)		Treated Virus Log_10_ TCID_50_/mL (qPCR Ct Value)		Not treated Virus Log_10_ TCID_50_/mL (qPCR Ct Value)		Treated Virus Log_10_ TCID_50_/mL (qPCR Ct Value)	
		Exp ^α^ I	Exp II	Exp I	Exp II	Exp I	Exp II	Exp I	Exp II
E30	2	3.00	3.25 (14.3)	0.00	0.00 (und *)	3.75	3.50 (14.3)	0.00	0.25 (17.99)
	10	3.50	3.75 (13.9)	0.00	0.00 (und)	3.25	4.00 (14.8)	0.00	0.00 (und)
	30	3.50	3.25 (14.2)	0.00	0.00 (17.6)	3.75	4.25(14.0)	0.00	0.00 (und)
	60	3.75	3.00 (14.2)	0.00	0.00 (und)	3.75	3.25 (und)	0.00	0.00 (14.0)
	120	3.50	3.25 (14.6)	0.00	0.00 (und)	4.25	3.75 (17.8)	0.00	0.00 (und)
RV SA11	2	5.00 (25.7)	5.00 (29.5)	2.25 (24.1)	1.25 (25.2)	4.75 (25.2)	5.00 (29.4)	1.25 (25.9)	1.25 (29.5)
	10	4.50 (24.9)	4.25 (28.7)	0.00 (36.6)	0.00 (38.4)	4.50 (25.6)	5.25 (28.2)	0.00 (27.6)	0.00 (36.9)
	30	4.25 (24.5)	4.00 (27.9)	0.00 (39.2)	0.00 (36.1)	4.75 (25.2)	5.00 (28.8)	0.00 (37.3)	0.00 (38.1)
	60	5.00 (24.5)	4.00 (28.6)	0.00 (29.2)	0.00 (36.0)	4.00 (25.5)	4.50 (27.2)	0.00 (28.6)	0.00 (37.8)
	120	4.50 (25.3)	3.50 (28.7)	0.00 (39.0)	0.00 (37.0)	4.50 (25.7)	4.25 (26.9)	0.00 (und)	0.00 (37.6)
HAdV2	2	3.75 (8.8)	5.00 (9.6)	0.00 (26.5)	0.75 (9.3)	4.50 (8.2)	5.00 (9.5)	0.00 (25.0)	0.00 (26.0)
	10	3.75 (7.7)	4.50 (8.9)	0.00 (27.2)	0.00 (11.9)	4.25 (7.4)	4.75 (9.2)	0.00 (23.6)	0.00 (24.5)
	30	3.75 (8.0)	4.50 (9.1)	0.00 (27.5)	0.00 (26.2)	3.50 (8.1)	5.00 (9.6)	0.00 (22.4)	0.00 (23.5)
	60	3.75 (8.2)	4.50(9.0)	0.00 (27.3)	0.00 (28.2)	3.75 (8.3)	3.75 (9.3)	0.00 (20.2)	0.00 (23.5)
	120	4.00 (7.4)	5.00 (8.9)	0.00 (26.9)	0.00 (24.6)	4.00 (9.3)	4.75 (9.8)	0.00 (22.8)	0.00 (22.6)

* = undetermined, α = experiment.

**Table 2 pathogens-12-01216-t002:** CT values of E30 in ultrafiltrated water samples treated with chlorine dioxide (pH 6.5–6.7) and chlorine (pH = 6.8) doses at 9.8 °C.

Time (min)	Chlorine				Chlorine Dioxide			
	Free Chlorine (mg/L)		CT (min·mg/L)		Chlorine Dioxide (mg/L)		CT (min·mg/L)	
	Exp I	Exp II	Exp I	Exp II	Exp I	Exp II	Exp I	Exp II
0	1.31	1.31			0.53	0.53		
2	0.37	0.50	1.79	1.81	0.20	0.26	0.73	0.79
10	0.17	0.29	3.95	4.97	0.26	0.18	2.57	2.55
30	0.15	0.21	7.15	9.97	0.17	0.26	6.87	6.95
60	-	0.11	-	14.77	0.11	0.13	11.07	12.80
120	0.12	0.11	13.00	21.37	0.07	0.09	16.47	19.40

**Table 3 pathogens-12-01216-t003:** CT values of RV SA11 in ultrafiltrated water samples treated with chlorine dioxide (pH 6.6–6.8) and chlorine (pH = 6.6–6.8) doses at 9.8 °C.

Time (min)	Chlorine				Chlorine Dioxide			
	Free Chlorine (mg/L)		CT (min·mg/L)		Chlorine Dioxide (mg/L)		CT (min·mg/L)	
	Exp I	Exp II	Exp I	Exp II	Exp I	Exp II	Exp I	Exp II
0	1.20	1.13			0.52	0.49		
2	0.41	0.31	1.61	1.44	0.25	0.16	0.80	0.80
10	0.25	0.26	4.25	3.72	0.21	0.29	2.64	2.60
30	0.20	0.18	8.75	8.12	0.09	0.19	5.64	7.40
60	0.14	0.14	13.85	12.92	0.09	0.12	8.34	12.05
118	0.10	0.11	21.05	20.42	0.03	0.06	11.94	17.45

**Table 4 pathogens-12-01216-t004:** CT values HAdV2 in ultrafiltrated water samples treated with chlorine dioxide (pH 6.4–6.7) and chlorine (pH 6.7–6.8) doses at 9.8 °C.

Time (min)	Chlorine				Chlorine Dioxide			
	Free Chlorine (mg/L)		CT (min·mg/L)		Chlorine Dioxide (mg/L)		CT (min·mg/L)	
	Exp I	Exp II	Exp I	Exp II	Exp I	Exp II	Exp I	Exp II
0	1.31	1.31			0.56	0.47		
2	0.55	0.35	1.86	1.66	0.25	0.35	0.85	0.85
10	0.32	0.21	5.34	2.24	0.22	0.24	2.73	3.21
30	0.15	0.14	10.04	5.74	0.16	0.21	6.53	7.71
60	0.10	0.12	13.79	9.64	0.10	0.21	10.43	14.01
120	0.08	0.13	19.19	17.14	0.03	0.08	14.33	22.71

## Data Availability

The data that support the findings of this study are available from the corresponding author upon request.

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
