# Peer review of "The Virucidal Effect of the Chlorination of Water at the Initial Phase of Disinfection May Be Underestimated If Contact Time Calculations Are Used"

_pathogens, 2023, doi:10.3390/pathogens12101216_

Round 1

Reviewer 1 Report

Saguti et al. present a manuscript describing the virucidal effect of water chlorination against three enteric viruses (echovirus, rotavirus, and adenovirus), all three extremely stable in the environment, and evaluate the reliability of the application of the CT method which is based on calculation of the ratio between the disinfectant concentration and the contact time, for estimation of the effectiveness of the disinfection procedure. 

Many viruses, especially those transmitted via the fecal-oral route, are quite stable and can survive for long periods of time in the aquatic environment, drinking water included. What is more, some of these viruses, like adenoviruses, are extremely resistant to UV radiation, the latter commonly used in drinking water treatment plants. On the other hand, concerns about by-products of chlorination have arisen and the current trend for water disinfection is less or even no chlorine treatment.

Human pathogenic viruses in drinking water are a global concern for public health. To prevent potential transmission by tap water, proper disinfection is essential based on a careful and detailed risk assessment of chlorination.

The CT method based on the calculation of the ratio between the disinfectant concentration and the contact time has its limitations, especially for shorter periods of time. These limitations are nicely illustrated by the results in the submitted manuscript.

To my opinion, the presented research work is relevant and significant. It is scientifically sound, well structured, methodologically balanced and concise, written in excellent English.

In fact, I have no major concerns. But I have some minor remarks and comments.

Here they follow:

The Title – I would advise avoiding using the abbreviation CT in the title. I am afraid that many potential readers will be initially confused because of the identity of the abbreviation with the more popular Cycle Threshold abbreviation (Ct).

The Introduction, lines 81, 82 – I would recommend a more detailed explanation of the correlation between the CT value and the log reduction.

Materials and Methods:

2.1. The Drinking Water Treatment Plant, line 104 – please explain the need to adjust the time by a factor of three.

2.2. Cl2 and ClO2 Stock Solution Preparation, line 108 – How much of the 12% stock solution of NaClO (volume in mL) was added to 4 L drinking water?

2.3. Assessing the Effect of Cell Culture Media on Cl2 and Clo2 Concentration in Ultra-Filtrated Water, lines 126, 127 – Describe the cell media for each of the cell cultures. Was it growth or maintenance media? Was fetal calf serum, antibiotics, L-glutamine, etc. added to the media? When was chlorine adsorption measured – before or after adding fetal serum to the media? Please explain in a more detailed manner. Please explain also the rationale for the dilutions 1/100 and 1/1000.

2.3. Assessing the Effect of Cell Culture Media on Cl2 and ClO2 Concentration in Ultra-Filtrated Water, line 127 – Please explain that this filtration is carried out after adding the disinfectant, which leads to the formation of organic debris that should be removed (as explained in Results). Otherwise, a question may arise in the reader if the water is already ultra-filtrated through filters with a nominal pore size of 20 nm (see line 100) why the need for another further filtration with a filter with larger pores?

2.5. Viral Infectivity Titers – I would recommend providing the titer of each of the virus stocks expressed as TCID50/mL. Thus it would be easily calculated by the potential reader how many TCID50 are there after adding 5 mL of virus stock in 5 L ultra-filtrated water and before adding the disinfectant.

Results:

Figure 1. I would recommend using one and the same scale for all the graphics. In the present figure, time is expressed by 20 min intervals for Cl2 doses, and by 30 min intervals for the ClO2 dose. If one and the same intervals were applied, comparison would be easier. Indicate the metrics for free chlorine on the ordinate. Are these mg/L?

3.2. Inactivation of viruses by chlorine and chlorine dioxide, line 225 – Please provide the virus titers of the virus stocks.

Table 1. Virus titers are expressed as TCID50 in a given volume unit, usually 1 mL (TCID50/mL). They should not be expressed as only TCID50. In addition, what do the figures for TCID50 mean? For example, what does 3.00 mean in Exp. I for E30, 2 min, not treated virus, and 3.25 for Exp. II, E30, 2 min, not treated virus? Are these 103.00 TCID50/mL and 103.25 TCID50/mL, respectively?

3.4. Rotavirus, line 267 – Titres are usually expressed as TCID50/mL. Please explain what is meant by 1/20 and 1/200. Dilutions?

Supplementary:

Table S2 – there is a typo mistake in “concentration” and another one, “media” at the beginning of the second sentence should be with a capital letter.

Table S3 – no need for capitalization of chlorine. TOC – please provide the abbreviation in full, although popular.

Table S4 – please provide the already introduced abbreviation for the rotavirus.

Reviewer 2 Report

The article is interesting and well written. It is read with great interest. Research has shown the important role of organic substances in water disinfection processes using both chlorine and chlorine dioxide. Chlorine consumption in the first 2 minutes of the process was high, which demonstrates the need to take into account the concentration of organic matter when determining the dose of the disinfectant. It was also shown that the biological activity of viruses should not be equated with their RNA residue in the solution.

I have some minor comments regarding the editing side of the article.

1. Fig.1 - units are missing on the vertical axes.

2. Table 1 - units are also missing.

3. Table 1 - the first digits of the numbers have been cut off from the right in columns 2, 3 and 4.

Last comment. When dosing NaClO into water, HClO and NaOH are formed, which naturally increases the pH of the water. Table S3 shows no significant changes in pH. Why? I think that the article needs a comment explaining this phenomenon, because I believe that Table S3 will not be attached to the article. This is important because the strong disinfectant HClO decomposes into OCl at higher pH, which is a much weaker disinfectant.
